# Chilblain-Like Lesions during COVID-19 Pandemic: The State of the Art

**DOI:** 10.3390/life11010023

**Published:** 2021-01-02

**Authors:** Andrea Bassi, Teresa Russo, Giuseppe Argenziano, Carlo Mazzatenta, Elisabetta Venturini, Iria Neri, Vincenzo Piccolo

**Affiliations:** 1UO Dermatologia Lucca-Azienda USL Toscana Nordovest, 55100 Lucca, Italy; bassi76@interfree.it (A.B.); carlo.mazzatenta@uslnordovest.toscana.it (C.M.); 2Dermatology Unit, University of Campania Luigi Vanvitelli, 80131 Naples, Italy; russo.teresa87@gmail.com (T.R.); g.argenziano@gmail.com (G.A.); 3Infectious Diseases Unit, Meyer Children’s University Hospital, 50139 Florence, Italy; elisabetta.venturini@meyer.it; 4Division of Dermatology, Department of Experimental, Diagnostic and Specialty Medicine, University of Bologna, 40138 Bologna, Italy; iria.neri@tin.it

**Keywords:** chilblain-like lesions, COVID-19, skin manifestation

## Abstract

SARS-CoV-2 infection has spread all over the world in the last year, causing millions of COVID-19 cases among humans with a large variability of symptoms and signs, including those on the skin. Among these, a contemporary cluster of chilblain-like lesions with no certain relationship with the infection has been reported. The aim of this paper is to delineate a profile of chilblain-like lesions and to establish the state-of-the-art knowledge about this new phenomenon.

## 1. Introduction

SARS-CoV-2 infection has spread all over the world in the last year, causing millions of COVID-19 cases among humans with a large variability of symptoms and signs, resulting in mild to very severe and sometimes lethal outcomes. Although respiratory symptoms are the most common, other findings could affect patients with COVID-19, including quite nonspecific cutaneous signs, such as urticarial, macular papular rash, and varicella-like eruptions. While the latter have been observed in patients with proven infection [1,2], during the first phases of the COVID-19 pandemic, a contemporary cluster of chilblain-like lesions with no certain relationship with the infection has been reported [3]. The aim of this paper is to delineate a profile of chilblain-like lesions and to establish the state-of-the-art knowledge about this new phenomenon.

## 2. Chilblain-Like Lesions: Definition and History

During the spring of 2020, when Europe was invaded by the coronavirus infection, a novel cutaneous sign was contemporarily observed by dermatologists, pediatricians, and general practitioners in different countries with variegate heterogeneity in its definition, ranging from chilblain-like lesions to pernio-like lesions or acroischemic lesions and others [4]. Hereinafter, we refer to these skin findings as chilblain-like lesions (CLLs). Although there is variability in the definition of this sign, common features were reported by all observers. First of all, CLLs mostly affected children and young people, in particular otherwise healthy adolescents, with no or mild symptoms, limited to painful or itchy sensations. Further, general symptoms were absent in most patients, even if some authors [5] reported patients with chilblains with associated suspicious symptoms of COVID-19 infection such as with fever, cough, dyspnea, and anosmia–ageusia. The morphology of the lesions also appeared identical to ordinary chilblains, mostly occurring on the feet (rarely hands) with acrocyanosis or cold toes with erythematous–edematous lesions occurring most frequently and, less frequently, bullous and necrotic lesions (Figure 1) [6,7]. The outbreak of these lesions was clearly “out-of-season”, as chilblains are usually related to cold and it was strange to see so many “spring” cases. Obviously, the first thought was to find a possible association with the COVID-19 infection. Due to the lockdown in most countries, it was initially difficult to test all affected patients, but according to the data reported in the literature, most of the testing for SARS-CoV-2, using real-time polymerase chain reaction (rt-PCR) on nasopharyngeal swabs, stool samples, and serum serologic analysis, did not reveal any association with the main infection in the majority of cases [5,8]. The few manuscripts which described a direct association with COVID-19 only involved sporadic situations where an entire family was infected with the virus and, consequently, also the children [6]. Next to the serologic testing for COVID-19, other standard serologic investigations were performed in most studies, including complete blood cell counts, liver and kidney function, prothrombin time, partial thromboplastin time, erythrocyte sedimentation rate, C-reactive protein levels, D-dimer values, antinuclear antibodies, antiphospholipid antibodies, hemolytic complement (C3, C4, CH50), cryoglobulinemia, parvovirus serology, and interferon-alpha (IFN-α) stimulation and detection [9]. Histology and immunofluorescence confirmed analogies with perniosis, sometimes associated with vasculitis and/or microthromboses [10,11,12]. In detail, two different patterns were detected from histopathology [12]: a chilblain-like histopathological pattern and a thrombotic vasculopathy pattern. In the first one (chilblain-like pattern), variable superficial to deep lymphocyte infiltrate around vessels was found. Additionally, the infiltrate was peri-eccrine in some patients and changes of eccrine glands were detectable in the majority of cases. The epidermal basal layer showed vacuolar alteration in most patients. Lichenoid interface dermatitis was occasionally found. The second pattern, namely, the thrombotic vasculopathy pattern, was characterized by no or slight infiltrate of inflammatory cells and many intraluminal fibrin thrombi associated with ischemic necrosis of epidermis. Moreover, dermoscopic investigations were performed. The typical erythematous–edematous lesions of toes mostly showed a prevalent coppery-red background along with dotted vessels and hemorrhagic dots, while dermoscopy of a blistering CLL revealed the same coppery-red background together with hemorrhagic dots and crusts. All these findings are not novel because they overlap with those we can normally find in pigmented purpuric dermatoses, especially the coppery-red background combined with hemorrhagic dots [13].

Definitely, all the previously mentioned features are typically found in chilblains and it is necessary to differentiate them with true acral ischemic lesions provoked by thrombotic vasculopathy due to the procoagulant state observed in critical, severely ill COVID-19 patients, for whom the direct effect of the virus may induce perivascular inflammation and secondary endothelial damage.

## 3. The State of the Art and Future Perspectives

Considering all of these common clinical, histological, and dermoscopic findings, the main doubt is the correlation between CLLs and COVID-19 [14,15]. Many manuscripts tried to speculate different possible direct associations with the virus, but to date, there are no reliable data about this. As we previously reported, the main elements which can justify a correlation with COVID-19 infection are the contemporaneity of CLL outbreak with the pandemic with the out-of-season appearance of this type of lesion, and secondarily, the positivity of anti-SARS-CoV or SARS-CoV-2 immunostaining on histological specimens of CLLs, which would confirm the direct detectability of the virus in the lesions [14,15]. According to some authors, in fact, endothelial damage induced by the viral particles could be the mechanism for the development of chilblains [16,17], even if these are only studies on a limited number of patients which should be confirmed by larger ones.

On the other hand, in most cases, rt-PCR and anti-SARS-CoV-2 serology showed negative results, with rare exceptions [18]. Therefore, negativity of rt-PCR on nasopharyngeal swabs may suggest that CLLs are not associated, with relative certainty, with COVID-19. Conversely, if rt-PCR was almost always negative at the time of CLL presentation, the SARS-CoV-2-specific IgA, IgM, and IgG antibody detection was described as positive in many reports [19,20], so authors concluded that CLLs may be associated with mild or asymptomatic SARS-CoV-2 infection and may be the expression of a late symptom of COVID-19. Obviously, the main limitation of these results is the differences in sensitivity/specificity between the tests that have been used. Moreover, some authors hypothesized a possible association with the cutaneous expression of type I interferon (IFN-I) [9,21]. In histopathological specimens obtained from patients with CLLs, an analysis of levels of IFN-I-induced proteins and signal transduction kinases was performed, showing that this type of IFN pathway was activated in cutaneous sections of patients with CLLs. As a consequence of that, it could be speculated that exposure to SARS-CoV-2 may induce local antiviral immune activation. The high concentration of IFN-I can likely be linked to a precocious control against the virus and to a mild and short course of disease. Moreover, antibody production may be suppressed by IFN-I, thus explaining the low rate of specific antibody positivity among patients with CLLs [22]. However, these patients did not show any other cutaneous or extracutaneous signs or symptoms typically present in other so-called “interferonopathies” observed in these patients. Interestingly, a strong difference in T-cell immunity has been demonstrated between patients with severe disease and those experiencing no symptoms or mild disease. In particular, while the former showed acute-phase SARS-CoV-2-specific T cells with a highly activated cytotoxic phenotype and, thus, seropositivity, the latter displayed T cells with a stem-like memory phenotype, providing robust protection against infection although seronegative. Thus, it could be speculated that patients with CLLs could represent a particular subset of asymptomatic COVID-19 cases, showing a strong immune T-cell response against the virus that could be interpreted as more cell-mediated cytotoxic. As a consequence of that, patients could develop cutaneous lesions and be seronegative because of the fast clearance of antibodies [23]. To conclude, a direct and sure association between CLL and COVID-19 is far from confirmed. Actually, we are led to think that the only certainty is that COVID-19 has completely changed our daily lifestyle, especially in the period from March to May where an almost total lockdown was present everywhere and many children spent their time at home, avoiding routine movements with a sedentary lifestyle, which could have led to the development of a kind of stasis capillaritis, as well as from bare-feet exposure to cool indoors. Obviously, we still need to understand why only a small number of these children developed these cutaneous lesions. These data could be confirmed by the total regression of the lesions during summertime and their reappearance during the second soft lockdown present in the period of November–December in many countries, even if no studies have been recently reported about the second wave of chilblain lesions. Only future studies with data on a larger series of patients with CLLs with systematic testing may definitely explain whether there is or not a real association.

## Figures and Tables

**Figure 1 life-11-00023-f001:**
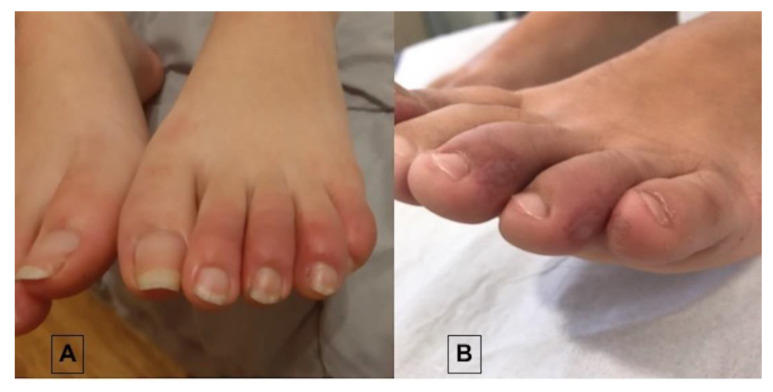
Erythematous–edematous (**A**) and bullous (**B**) chilblain-like lesions: (**A**) 13-year-old male, COVID-19 negative; he had spontaneous resolution in one month; (**B**) 12-year-old male, COVID-19 negative; resolution was achieved with daily application of high-potency steroids.

## Data Availability

Not applicable.

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
