# Peer review of "Chilblain-Like Lesions during COVID-19 Pandemic: The State of the Art"

_life, 2021, doi:10.3390/life11010023_

Round 1

Reviewer 1 Report

The authors present an interesting explanation about chilblain-like lesions during COVID-19 pandemic.

I would recommend acceptance and publication.

Author Response

Thank you for your comments and appreciation.

Reviewer 2 Report

This short opinion paper discusses the relationship between SARS-Cov-2 and chilblain-like lesions (CLL) which appeared during the COVID-19 pandemic. It is not very clear how authors reviewed the relevant literature, as some relevant papers on the topic are missing from their list of references (see: Br J Dermatol. 2020;183:866; J Eur Acad Dermatol Venereol 2020 Jun 27:10.1111/jdv.16779. doi: 10.1111/jdv.16779; J Am Acad Dermatol 2020;83:870; Med Hypothesis https://doi.org/10.1016/j.mehy.2020.109959). Much of what is discussed in their paper has already been reported before, at least party, in these previous papers (and several others from the literature).

Concerning the content of the review, I would suggest the authors to discuss more the following points:

- the detection of SARS-CoV-2 with immunohistochemistry and electron microscopy in skin lesions from one group of authors; although (in my view) these results are somewhat questionable, they need to be more extensively commented

- the lower (reported?) incidence of CLL during the second wave of COVID-19 compared with the first one.

The English needs editing as it often sounds awkward and contains some grammatical and syntax errors that occasionally detract from reading. Examples: the CLL were first observed not by ‘researchers’ but by practicing dermatologists and general practitioners. ‘Poor’ symptoms should read ‘mild’ symptoms. ‘identical with CLL’ should read ‘identical with ordinary chilblains’.

References 8 and 20 are one and the same!

J Am Acad Dermatol. 2020 Sep;83(3):870-875

Author Response

  • This short opinion paper discusses the relationship between SARS-Cov-2 and chilblain-like lesions (CLL) which appeared during the COVID-19 pandemic. It is not very clear how authors reviewed the relevant literature, as some relevant papers on the topic are missing from their list of references (see: Br J Dermatol. 2020;183:866; J Eur Acad Dermatol Venereol 2020 Jun 27:10.1111/jdv.16779. doi: 10.1111/jdv.16779; J Am Acad Dermatol 2020;83:870;

Med Hypothesis https://doi.org/10.1016/j.mehy.2020.109959). Much of what is discussed in their paper has already been reported before, at least party, in these previous papers (and several others from the literature).

We have added thees 4 manuscripts in the references

  • “the detection of SARS-CoV-2 with immunohistochemistry and electron microscopy in skin lesions from one group of authors; although (in my view) these results are somewhat questionable, they need to be more extensively commented”

We have added a sentence and a new reference. There are only few studies which describe the presence of the viral particles in the endothelial cells of histological samples, so according to us it can not be considered a certainty. Moreover, in all these studies the rt-PCR detection of the virus was negative

  • “the lower (reported?) incidence of CLL during the second wave of COVID-19 compared with the first one”

No reports about the second wave of chilblains are present in the literature, in fact this is only a personal opinion about our Italian experience due to the second soft lockdown present in our country compared to the March-April general lockdown. Obviously, these data need to be documented by new studies. We have added a sentence

  • “The English needs editing as it often sounds awkward and contains some grammatical and syntax errors that occasionally detract from reading. Examples: the CLL were first observed not by ‘researchers’ but by practicing dermatologists and general practitioners. ‘Poor’ symptoms should read ‘mild’ symptoms. ‘identical with CLL’ should read ‘identical with ordinary chilblains’.”

We have improved the language and corrected the text according the reviewer suggestions.

  • “References 8 and 20 are one and the same!”

We have deleted one of these references

Reviewer 3 Report

The article tries to establish the state art of knowledge about the chilblain-like lesions, a constantly evolving issue in the covid-19 pandemic.  New evidence are emerging leading to not unequivocal conclusions. The article summarize properly the evidence but some statements should be stated.

Lines 56-63: You assume two main histological patterns were observed, chilblain-like and trombotic vasculopathy pattern, please specify if it is possible to consider this patterns as clearly divided or not with proper references.

Lines 83-95: You assume CLL may be associated with mild or asymptomatic Sars-CoV-2 infection. Please consider the publication on the British Journal of Dermatology “Most Chilblains Observed During the COVID-19 Outbreak Occur in Patients Who Are Negative for COVID-19 on Polymerase Chain Reaction and Serology Testing” L. Le Cleach et al.; include and comment the considerations emerging in the article.

Lines 111-113: interesting consideration, please verify in literature the necessary evidence to sustain it.

Lines 116-117: You identify the role of lifestyle changes as an important element to consider in the association between skin lesions and viral infection. Please deepen this debated issue considering the different viewpoints in literature.

Lines 45-48: The observation…Covid-19 infection. Please, rephrase.

Figure 1: please, specify which kind of patients are referred to (age, sex, covid-19 positive or suspected, clinical course).

Author Response

  • “You assume two main histological patterns were observed, chilblain-like and trombotic vasculopathy pattern, please specify if it is possible to consider this patterns as clearly divided or not with proper references.”

We have added the reference in the text: Sohier P et al.

  • “You assume CLL may be associated with mild or asymptomatic Sars-CoV-2 infection. Please consider the publication on the British Journal of Dermatology “Most Chilblains Observed During the COVID-19 Outbreak Occur in Patients Who Are Negative for COVID-19 on Polymerase Chain Reaction and Serology Testing” L. Le Cleach et al.; include and comment the considerations emerging in the article.”

We have added a sentence in the text discussing the possible systemic associated symptoms

  • “You identify the role of lifestyle changes as an important element to consider in the association between skin lesions and viral infection. Please deepen this debated issue considering the different viewpoints in literature.”

We have added a sentence in the text discussing this

  • Lines 45-48: The observation…Covid-19 infection. Please, rephrase

We have rephrased the sentence in the text

  • Figure 1: please, specify which kind of patients are referred to (age, sex, covid-19 positive or suspected, clinical course).

We have added information in the figure legend